# Evaluation of the Clinical Use of Ceftriaxone among In-Patients in Selected Health Facilities in Uganda

**DOI:** 10.3390/antibiotics10070779

**Published:** 2021-06-25

**Authors:** Paul Kutyabami, Edson Ireeta Munanura, Rajab Kalidi, Sulah Balikuna, Margaret Ndagire, Bruhan Kaggwa, Winnie Nambatya, Pakoyo Fadhiru Kamba, Allan Musiimenta, Diana Nakitto Kesi, Victoria Nambasa, Allan Serwanga, Helen Byomire Ndagije

**Affiliations:** 1Department of Pharmacy, School of Health Sciences, College of Health Sciences, Makerere University, University Rd, 10218 Kampala, Uganda; paulkutyabami72@gmail.com (P.K.); kalidi.rajab@mak.ac.ug (R.K.); baliksh@gmail.com (S.B.); nmargic@gmail.com (M.N.); bruhan.kaggwa@mak.ac.ug (B.K.); winnie.nambatya@mak.ac.ug (W.N.); pakoyo.kamba@mak.ac.ug (P.F.K.); 2Department of Statistical Methods and Actuarial Sciences, School of Statistics and Planning, College of Business and Management Sciences, Makerere University, University Rd, 10218 Kampala, Uganda; musiimentaallan123@gmail.com; 3Directorate of Product Safety, National Drug Authority, Lumumba Avenue, 10106 Kampala, Uganda; dnakitto@nda.or.ug (D.N.K.); vnambasa@nda.or.ug (V.N.); aserwanga@nda.or.ug (A.S.); hndagije@nda.or.ug (H.B.N.)

**Keywords:** antimicrobials, ceftriaxone, drug utilization review

## Abstract

Ceftriaxone has a high propensity for misuse because of its high rate of utilization. In this study, we aimed at assessing the appropriateness of the clinical utilization of ceftriaxone in nine health facilities in Uganda. Using the World Health Organization (WHO) Drug Use Evaluation indicators, we reviewed a systematic sample of 885 patients’ treatment records selected over a three (3)-month period. Our results showed that prescriptions were written mostly by medical officers at 53.3% (470/882). Ceftriaxone was prescribed mainly for surgical prophylaxis at 25.3% (154/609), respiratory tract infections at 17% (104/609), and sepsis at 11% (67/609), as well as for non-recommended indications such as malaria at 7% (43/609) and anemia at 8% (49/609). Ceftriaxone was mostly prescribed once daily (92.3%; 817/885), as a 2 g dose (50.1%; 443/885), and for 5 days (41%; 363/885). The average score of inappropriate use of ceftriaxone in the eight indicators was 32.1%. Only 58.3% (516/885) of the ceftriaxone doses prescribed were administered to completion. Complete blood count and culture and sensitivity testing rates were 38.8% (343/885) and 1.13% (10/885), respectively. Over 85.4% (756/885) of the patients improved and were discharged. Factors associated with appropriate ceftriaxone use were gender, pregnancy status, days of hospitalization, health facility level of care, health facility type, and type of prescriber.

## 1. Introduction

Antibiotics are among the most commonly prescribed drug classes in developing countries such as Uganda. The rates of antibiotic prescriptions have greatly risen and are reported to be double beyond the World Health Organization (WHO) recommendation of 30%, especially in low- and middle-income countries (LMICs) [1,2]. The most prescribed antibiotics are the cephalosporins [3,4,5,6].

Ceftriaxone is a third-generation cephalosporin, beta-lactam antibiotic that is administered intravenously or intramuscularly for a broad range of susceptible infections. It is highly efficacious and largely safe, which underlies its popularity in clinical use in many parts of the world [3,7]. For instance, ceftriaxone is recommended for the treatment of over 30 conditions in the current Uganda Standard Treatment Guidelines (STGs) [8]. It is the most commonly prescribed antibiotic for patients in hospitals, with over 80% exposure among all in-patients admitted to Mulago National Referral Hospital, Uganda [9]. Findings from other studies in Spain, Ethiopia, Saudi Arabia, and South Korea have provided similar outlooks on ceftriaxone utilization [3,6,7,10]. Ceftriaxone is widely used in various countries for majorly surgical prophylaxis and in the management of respiratory tract infections (especially pneumonia) and urinary tract infections [6,7,10,11,12,13,14].

Ceftriaxone, like most cephalosporins, has a high prevalence of inappropriate prescriptions [3,4,5,6]. Ceftriaxone has a high propensity for misuse because it is utilized in high quantities clinically and is prescribed in an uncontrolled manner in many countries including Uganda [3,9]. Inappropriate use of ceftriaxone is reported to occur in over 34–70% of cases [3,4,10]. Furthermore, empiric use of ceftriaxone is estimated to be as high as 80–90% in hospitalized patients in some countries [6,15]. A cross-sectional survey previously carried out in Uganda at Mubende Regional Referral hospital on clinical utilization of ceftriaxone reported 81% inappropriate administration and no culture and sensitivity testing in over 93% of the patients sampled [13]. Similar findings have been reported in studies carried out in other countries in Africa, Middle East, and Asia [3,7,10,12,14,16,17,18].

Inappropriate use of antibiotics such as ceftriaxone accelerates the emergence of antimicrobial resistance, increases costs of treatment, affects productivity, and exposes patients to unnecessary side effects, and can also result in death [19,20,21,22,23]. Death rates are higher in surgical cases, especially patients who have undergone maternal cesarean section in low- and middle-income countries. The high mortality in this group is attributed to the development of post-surgical site infections that occur in 2–24% of patients, mainly as a result of inappropriate antibiotics use [23,24,25,26]. For instance, patients missing post-cesarean section antibiotic doses are 2.5 times more at risk of obtaining post-surgical site infections, and these infections can lead to death in over 5.3% of such patients [23].

There are emerging reports of the ineffectiveness of certain antibiotics including ceftriaxone worldwide. This ineffectiveness is largely due to the escalation of antimicrobial resistance accelerated by inappropriate prescription and ineffective antimicrobial stewardship [21,27]. Non-susceptible organisms cause higher rates of morbidity and mortality and significantly cost more to be treated than the susceptible organisms [3]. Worldwide, resistance to third-generation cephalosporins such as ceftriaxone is estimated to average 12% and 36% for methicillin-resistant *S. aureus* and *E. coli*, respectively [20]. In East Africa, resistance to ceftriaxone among Gram-negative infections is reported to be as high as 24–69% [28,29]. In Uganda, for instance, multiple strains of *Klebsiella pneumoniae* and *E. coli* were reported to be non-susceptible to ceftriaxone in culture and sensitivity studies performed with rates of 85% and 15%, respectively [30]. This pattern has causally been associated with several factors, most profoundly irrational use of antibiotics (overuse, misuse) and supposed low-quality antibiotics [3,22].

Several anecdote reports in Uganda blame the ineffectiveness of ceftriaxone on the quality of the commercially available brands on the market rather than inappropriate use. These were, however, dismissed by preliminary studies done by the National Drug Authority (NDA) on the quality of brands available on the Ugandan market that reported their conformity to pharmacopoeial standards adopted for Uganda. Ceftriaxone is in many ways an unrestricted antibiotic, and such atypical use of any antibiotic encourages the emergence of non-susceptible organisms. It can therefore be inferred that the reported cases of the inefficacy of ceftriaxone may be the result of this relationship. In the advent of these findings, a thorough assessment of the utilization of ceftriaxone in clinical settings in Uganda was the next logical step in attempting to understand the reasons for the reported inefficacy of ceftriaxone. In this study, therefore, we conducted a drug utilization evaluation (DUE) in nine (9) health facilities in Uganda to obtain a wider perspective.

## 2. Results

### 2.1. Social Demographic Characteristics

We systematically selected and assessed a total of 885 patients’ records from nine health facilities selected from different regions of Uganda, and at different levels of care and affiliation. The age of the patients whose files were reviewed ranged from 0 to 98 years (mean age + SD; 27.2 ± 22.7). The weight ranged from 2.4 to 95 kg (mean weight ± SD; 35.0 ± 26.1). Medical officers wrote most of the prescriptions (53.3%; 470/882), followed by clinical officers (13.9%; 123/882) (Table 1). Specialists made most of the prescriptions in the private for-profit (PFP) facility, accounting for 60.6% of the ceftriaxone prescriptions found in this facility (60.6%; 63/104). The biggest number of prescribers in both public and private not-for-profit (PNFP) were medical officers (Figure 1).

### 2.2. Appropriateness of Ceftriaxone Use

The average percentage scores for the nine (9) health facilities based on facility type categorization were as follows: appropriate indication—83.3% (±5.6), appropriate dose—88.8% (±3.2), appropriate duration—84.1% (±5.7), laboratory testing performed—62.2% (±28.4), prescribing by generic name—87.1% (±21.7), presence of drug interactions—4.1% (± 4.1), and appropriate dispensing—58.2% (±26.3). The WHO targets for these same parameters are 90%, 95%, 95%, 100%, 100%, 0%, and 95%, respectively [31] (Table 2). General hospitals (GH) scored highest overall in the appropriateness of indication (85%), dose (90%), and duration (86.2%). Health center IVs (HCIV) scored highest overall regarding the appropriateness of dispensing (79.6%) and generic prescribing (100%). The regional referral facility (RRH) had better laboratory testing rates (84.7%), while HCIVs had the lowest (36.2%). The overall average percentage score in the eight (8) indicators for the nine (9) health facilities was 67.9%. The most common indications for which ceftriaxone was prescribed were surgical prophylaxis—25.3% (154/609) (trauma, labor, and pregnancy complications), respiratory tract infections (RTIs) (majorly pneumonia)—17% (104/609), sepsis—11% (67/609), and gastrointestinal tract (GIT) infections—9.5% (58/609). Ceftriaxone was found to be prescribed for conditions not listed under the STGs, for instance, malaria—7% (43/609) and anemia—8% (49/609) (Figure 2). For some patients (1.8%) (16/885), no indication for ceftriaxone prescription was given.

Only 58.3% (516/885) of the ceftriaxone doses prescribed were administered to completion (Table 3). On various occasions, doses were not administered on some days with the highest doses missed being on day 5 (166 doses) and day 3 (151) (Figure 3). Some patients (1.4%; 12/882) had the dosing frequency of ceftriaxone changed from once a day to twice a day, contrary to the prescription. Furthermore, during sampling of prescription to select out those who were prescribed ceftriaxone, over 15 patient treatment records (1.7%) were found to have ceftriaxone administered, yet it was not on the prescription. Various reasons were given for the failure to have all the prescribed doses of ceftriaxone administered accordingly. These include the patient being referred for further management (3.6%; 11/303), patient’s condition improved and as such patient was discharged (19.1%; 58/303), ceftriaxone switched to another drug (20.1%; 61/303), and patient having died (12.5%; 38/303), among others. However, 19.1% of the patients missed the doses because either the drug was out of stock (11.2%; 34/303) or because of self-discharge (7.9%; 24/303). A total of 24.4% (74/303) of the patients who did not complete the doses of ceftriaxone prescribed had no reason provided to explain this.

One or more potential drug interactions with ceftriaxone were identified on the prescriptions reviewed for eight out of the nine health facilities. The overall average prevalence of possible drug interactions was 4% (range: 0–15%). The interacting drugs on the prescriptions containing ceftriaxone were furosemide and calcium-containing products, e.g., Calcivita^®^ and Ringer’s lactate.

In this study, CBC and culture and sensitivity tests were performed in only 38.8% (343/885) and 1.13% (10/885) of the patients, respectively. Other tests performed included liver functional test (LFTs) and renal functional test (RFTs), widal tests, and TB test. Furthermore, 38.1% (337/885) of the patients were prescribed and treated with ceftriaxone without any laboratory test performed (Figure 4).

### 2.3. Ceftriaxone Doses, Dose Frequencies, and Dose Durations Prescribed

The most prescribed dose of ceftriaxone in all the health facilities was 2 g (50.1%; 443/885). The 2g ceftriaxone dose was the most prescribed dose under both categories of level of care and facility type (Table 4). Medical officers, clinical officers, and specialists preferably prescribed the 2 g dose, i.e., 57.2% (269/470), 43.9% (54/123), and 55.8% (58/104), respectively, while the nurses preferably prescribed the 1g dose (36.6%; 30/82) (Table 5). The most prescribed dosing frequency was once daily (92.3%; 817/885). The average duration that ceftriaxone prescribed was 3.87 (SD = 2.28). The most prescribed duration for the administration of ceftriaxone was 5 days (41.0%; 363/885), followed by 3 days (29.9%, 265/885). Clinical officers and medical officers mostly prescribed the patients ceftriaxone for 5 days, i.e., 57.7% (71/123) and 47% (221/470), respectively, while nurses and specialists prescribed most patients Ceftriaxone for administration for 3 days (31.7%; 33/104, and 31.7%; 26/82, respectively). Overall, 5.31% (47/885) of the patients were prescribed ceftriaxone without indicating the duration of administration/use. This was mostly performed by medical interns, with 22.9% (22/96) of their patients’ prescriptions having no specified duration of ceftriaxone use.

### 2.4. Treatment Outcomes

The duration patients were admitted ranged from 1 to 49 days (mean 5, SD 4.8 days). Most of the patients (85.6%; 756/885) in the nine HFs improved upon treatment with ceftriaxone and were successfully discharged, while 3.8% (34/885) were referred for further management. However, 10.6% of the patients did not have the desired outcome, with 5.6% (49/885) having died. The PFP HF had the biggest proportion of patients discharged (97%; 97/100). Public HFs had the largest proportion of self-discharge cases (6.8%; 40/587), while PNFP HFs had the most deaths registered (13.6%; 27/198) (Figure 5).

### 2.5. Factors Associated with Appropriate Ceftriaxone Use

At bivariate analysis, the gender of respondents, pregnancy status, days of hospitalization, facility level of care, facility type, and prescriber were statistically significant. When we controlled for confounders at the multivariate stage, the gender of respondent, pregnancy status, days of hospitalization, facility level of care, facility type, and prescriber remained statistically significant. General hospitals and HCIVs were 3.4 and 3.6 times, respectively, as likely as regional referral hospitals to use ceftriaxone appropriately (*p* = 0.002; *p* = 0.001). PFP facility was 7.4 times as likely as government health facilities to use ceftriaxone appropriately (AOR = 7.4; 3.5–15.8, *p* = 0.000). Appropriate clinical usage of ceftriaxone was 60% less likely among pregnant patients than those that were not (AOR = 0.4; 0.2–0.8, *p* = 0.006), and 80% less likely among females than in males (AOR = 0.2; 0.1–1.0, *p* = 0.045) (Table 6).

## 3. Discussion

Our study aimed at evaluating the clinical use of ceftriaxone among in-patients in selected health facilities in Uganda based on eight (8) of the WHO’s recommended guidelines, and the overall appropriate use indicators were all below those recommended by the WHO. The WHO recommends the following indicators for drug utilization evaluation studies: indication, dose, duration, laboratory investigations, appropriate combination therapy, accurate dispensing, proper discontinuation, generic prescribing, proper continuation of therapy, and treatment outcomes [32]. We used eight of these in our study, namely; indication, dose, duration, laboratory investigations, accurate dispensing, drug-drug interactions, generic prescribing, and treatment outcomes. The most common indications for prescribing ceftriaxone were surgical prophylaxis (trauma, labor, and pregnancy complications), RTIs (majorly pneumonia), sepsis, and GIT infections, and these were largely in agreement with those obtained in other studies performed on the African continent and beyond [6,7,10,11,13,14,15]. However, in our study, surgical prophylaxis was a more common indication for which ceftriaxone was prescribed. The most prescribed dose of ceftriaxone was 2g, as was also previously reported in other related studies in Uganda, Eritrea, Saudi Arabia, and Ethiopia [6,10,12,13,17]. The ceftriaxone dose recommended ranges from 1 to 2 g in adults and 50 to 100 mg/kg in children [8]. The recommended dosing frequency for ceftriaxone varies from once daily to twice daily, depending on the condition, and the duration is 1 day to over 21 days, depending on treatment response [8,33]. The most prescribed frequency for the administration of ceftriaxone in this study was once daily (92.3%; 817/885), which was in agreement with similar studies in Tanzania [11], Ghana [34], and Saudi Arabia [6], and also in Uganda, which reported 100% for once-daily administration [13]. However, in a related study carried out in Ethiopia, twice-daily dosing was the most prescribed [7]. The mean duration (3.87 days) over which ceftriaxone was prescribed in our study was similar to that reported in a related study conducted at one hospital in Uganda [13]. Studies in Ethiopia and Eritrea reported a higher mean duration over which ceftriaxone was prescribed; i.e., 11.47 days [14], 5.6 days [17], 5.2 days [10], and 6.79 days [12]. In another related study carried out in 10 university hospitals in Korea, the mean duration prescribed was 10.3 days [3]. The difference in the duration over which ceftriaxone is prescribed could be explained by the fact that these other studies were carried out at only higher tier facilities (regional referral and specialist hospitals), while our study involved both lower- and higher-tier health facilities. Higher-tier HFs usually manage referral patients, and most of these usually have conditions in more advanced stages that require longer treatment durations.

In this study, we found appropriate use significantly determined by gender, pregnancy status, days of hospitalization, facility level of care, facility type, and prescriber category. General hospitals (GH) under the level of care and private for-profit (PFP) under the type of facility had better ceftriaxone clinical use practices. Age was found not to be a significant determinant of the appropriateness of the use of ceftriaxone just like in other studies [7,14]. However, in these same studies, gender was reported as not being a significant determinant of the appropriateness of use of ceftriaxone, unlike in our study, wherein the appropriateness of ceftriaxone use was significantly better in male patients. Furthermore, our study found more rational clinical use of ceftriaxone among non-pregnant females as compared to pregnant ones. Just like in related studies [7,14], days of hospitalization were found as a significant determinant of the appropriateness of the use of ceftriaxone. In our study, inappropriate ceftriaxone use was found to be more likely in patients admitted for shorter periods (≤3 days). Furthermore, ceftriaxone prescriptions by intern doctors were more likely to be inappropriate when compared to those of most of the other prescribers. This could be due to the inadequate experience of intern doctors with the use of the STGs and possibly a desire to explore various antibiotics, including ceftriaxone. In this study, inappropriate ceftriaxone prescription by intern doctors was attributed to issues such as not specifying the duration of use of ceftriaxone, since over 22.9% of the prescriptions made by intern doctors lacked this.

The overall inappropriate use score in our study was 32.1% and was lower than that reported in related studies in other countries. Inappropriate prescription of ceftriaxone has been determined to be at 39.4–87.9% in Ethiopia [7,10,14,17], 62.4% in Eritrea [12], 53% in the USA [19], and 34.5% in Korea [3]. The differences in the score for inappropriate use could be attributed to factors such as country differences in the STGs, differences in prescribers’ qualifications and experiences, the extent of training of healthcare workers on the use of STGs, and the assessment protocols used. Inappropriate ceftriaxone use was largely due to inappropriate dispensing, inadequate laboratory testing, wrong indications, wrong doses, and wrong dose duration. Ceftriaxone was used in over 37.8% of the patients without any laboratory test performed. Furthermore, 61.2% and 98.9% of patients lacked CBC and culture and sensitivity test results, respectively. In related studies, culture and sensitivity testing was not conducted in; 93% of patients in Uganda [13], 53.2–68.7% in Ethiopia [7,14], 91.1% in Sudan [15], and 33.5% in Korea [3]. CBC and culture and sensitivity tests are important in assessing treatment outcomes; for instance, a CBC value in normal limits (3.7–9.4 × 10^9^ mm^3^) [35] while culture and sensitivity testing facilitates the selection of the most appropriate antibiotics [27,36,37]. In LMICs, the capacity of clinical microbiology laboratories is very low, and even where capacity is not limited, such laboratories are underutilized [22,38,39]. The inadequate capacity is majorly due to inadequate laboratory infrastructure, lack of adequately trained/qualified staff, and limited resources to procure laboratory consumables [40,41,42]. In Uganda, a shortage of laboratory technologists exists in health facilities, with only 50% of posts for these filled at general hospitals. Furthermore, only 35.7% and 35.7% of laboratory technologist positions are filled at non-hub and hub hospital laboratories [42]. Most physicians prefer not to request culture and drug sensitivity testing due to the related high costs and time delays [12,14,36]. The high prevalence of lack of laboratory testing is an indicator of poor patient care management systems at these health facilities [16]. Lack of microbiology laboratory testing escalates misdiagnosis and can result in power management of life-threatening conditions such as post-surgical site infections [23,25]. The challenges above call for more focus on support towards laboratory infrastructure, consumables, and laboratory human resources in line with the Antimicrobial Resistance National Action plan [27].

The overall average inappropriate indication in this study was 16.7% and was lower than related retrospective studies carried out in Eritrea at 44% [12] and Ethiopia at 39.4% [10], as well as a prospective study in Ethiopia that had a score of 18.5%. It was, however, higher than that reported from other retrospective studies in Ethiopia at 4.7% [7,14]. Inappropriate indication in our study was attributed to some patients being treated with no diagnosis documented, as well as ceftriaxone being prescribed for non-recommended conditions. Ceftriaxone was also found to be prescribed for conditions not listed under the standard treatment guidelines [8], for instance; malaria, anemia, peptic ulcer disease, pulmonary TB, and hypertension, among others. The drivers for this could be a lack of proper diagnostics infrastructure [36,41], deficiencies in knowledge among prescribers, or strive for monetary gains. Over 1.8% of patients were prescribed ceftriaxone with no diagnosis indicated. This was, however, much better than the reported prevalence from related studies in Ethiopia [7,16], Sudan [15], and Tanzania [18]. Prescribing a drug without including the diagnosis invalidates a prescription, and such a prescription should not be dispensed and administered to the patient since it makes monitoring of patients’ progress hard.

Furthermore, inappropriate utilization as regards prescribing correct doses and dose durations were 11.2% and 15.9%, respectively. The inappropriate dosing score was lower than the 45% reported in a related study in Eritrea [12]. In addition, the score for the inappropriate duration was lower than the reported 47–71% scores in related studies in Ethiopia and Eritrea [12,14,16,17]. Most cases of inappropriate use are always due to inappropriate duration, unlike in our study where inappropriate indication was most prevalent. Injectable antibiotics such as ceftriaxone are recommended mostly for the initial management of severe infections and are they are supposed to be replaced with other alternatives when the patient’s condition improves. However, standard treatment guidelines specify given dosing frequencies and durations for some specific conditions. For instance, the UCG (2016) recommends at least 5–7 days for pneumonia, 14 days for pyelonephritis, 10–14 days for meningitis and typhoid, and 7–10 days for acute chest syndrome and acute abdomen in sickle cell disease (SCD) among others [8].

Over 85.6% of the patients were successfully discharged as having completely been treated. This figure was comparable to that reported in a previous study carried out by Manirakiza et al. (2019) in Uganda. However, the high rate of patients getting better cannot be directly attributed to the appropriateness of the use of ceftriaxone. The dismal rates of laboratory testing, with culture and sensitivity as low as 1.1% could not enable us to obtain generalizable results to make such conclusions. In our study, 5.6% of the patients were reported to have died, and this could be attributed to probably treatment failure, co-morbidities, or improper diagnosis. The possibility of occurrence of treatment failure could be justified by the fact that a very high proportion (41.7%; 369/885) of the ceftriaxone doses prescribed were not completed; culture and sensitivity testing rates were very low; and drug administration was irregular, with over 650 doses missed during the period of treatment. A related study in Uganda reported that only 18% of patients completed the prescribed doses of ceftriaxone and had regular administration [13]. In our study, the overall score for possible interactions was 4%, and this was comparable to a related study in Sudan [15]. Ringer’s lactate (contains calcium gluconate) was found to be prescribed alongside ceftriaxone in some cases, yet co-administration of ceftriaxone with calcium-containing products is reported to result in occasional occurrences of possible or probable embolic events [43]. Furthermore, furosemide was also being co-administered with ceftriaxone, yet this is reported as having the potential to worsen kidney function [44]. Co-administration of ceftriaxone with furosemide or Ringer’s lactate was also reported as being prevalent in related studies carried out in Ethiopia [10] and Sudan [15].

## 4. Materials and Methods

### 4.1. Study Setting and Design

We conducted a document review study in nine (9) healthcare facilities at different; levels of care (general hospital, regional referral, health center IV), geographical locations (central, eastern, northern, and western), affiliation, and ownership (government, private for-profit, and private not-for-profit) to assess clinical utilization of ceftriaxone among in-patients. We reviewed records for at least three (3) months from the time the study was conducted. The records we reviewed included patients’ prescriptions, in-patient registers, treatment sheets, and laboratory registers. The facilities were proportionately selected according to their total number in the country [45].

### 4.2. Sample Size Determination and Sampling Procedure

We set the sample size at 100 patient records per health facility based on the WHO recommendation of at least 75 prescriptions per health facility for DUE studies [31]. We systematically sampled, selected, and reviewed patient medical records between November 2019 and January 2020. With the help of the records officers at the facilities, the total number of patient records that satisfied the selection criteria were obtained and a sampling frame (K) was determined. Then, every Kth patient record was picked and reviewed until the required sample size was obtained. Due consideration was given to facilities with different wards by proportionately selecting records from the different wards.

### 4.3. Data Collection

We adapted a standard structured data collection tool of the World Health Organization (WHO) for collecting data. We pretested the tool at the acute care unit of Mulago Hospital. The tool covered the following parameters: patient demographics, prescriber qualification, indication for use, the dose, frequency, duration, contraindication, interactions, culture, and sensitivity testing, as well as treatment outcomes such as length of stay, adverse drug reactions, treatment failure/switch, and cure/death. Data from the sampled patients’ files were reviewed, retrieved, and entered into the data abstraction sheet with the help of two facility staff under the supervision of one of the co-investigators.

### 4.4. Data Handling and Analysis

We entered the data in Epi-Data, cleaned it, and exported it to Stata/SE 14 for analysis. Continuous variables such as age and weight were summarized as means, standard deviations, and ranges, while percentages and proportions were used for categorical variables. We evaluated the prescribing patterns for ceftriaxone using the percentage of ceftriaxone that was prescribed, dispensed, and administered to the patient appropriately. We then evaluated the prescriptions against the current Uganda Standard Treatment guidelines (UCG, 2016) to assess compliance, and we presented these results as percentages and summarized the treatment outcomes following the use of ceftriaxone as percentages. We analyzed appropriateness for eight indicators using a WHO-adapted tool. We conducted logistic regression to determine factors associated with inappropriate use of ceftriaxone. Associations with a significance level of less than 0.2 (*p* ≤ 0.2) at the bivariate analysis were included in the multivariate logistic regression analysis to adjust for any confounding and reported as adjusted odds ratios at the 95% significance level, *p* ≤ 0.05.

## 5. Conclusions

Our study revealed that inappropriate use of ceftriaxone is high. This inappropriate use could result in increased resistance and more complicated management of infections, which could lead to death and catastrophic expenditures, let alone burden the health system as a whole. Inadequacy of laboratory testing, inadequate dispensing of prescribed doses to patients, and prescribing for wrong indications at the nine health facilities selected across Uganda contributed to high inappropriate use of ceftriaxone. Ceftriaxone is being prescribed largely empirically and for surgical prophylaxis and is to a great extent being prescribed for conditions not provided for as per the current Uganda Clinical Guidelines. Furthermore, culture and sensitivity testing rates are extremely low, and there is a very high prevalence of missed doses during the period of hospitalization. Considering the above, the increase in reports of purported reduced efficacy of ceftriaxone could be attributed to its irrational use in clinical settings across Uganda. The implementation of the Uganda Antimicrobial Resistance National Action Plan (AMR-NAP), 2018–2023, needs to be reinforced to realize its objective of promoting optimal access and use of antimicrobials. This should be performed through its proposed interventions such as strengthening laboratory infrastructure, improvement of laboratory human resources, establishing and strengthening medicine therapeutics committees, and incorporating antimicrobial stewardship courses in pre-service and in-service curricula of health professionals.

## Figures and Tables

**Figure 1 antibiotics-10-00779-f001:**
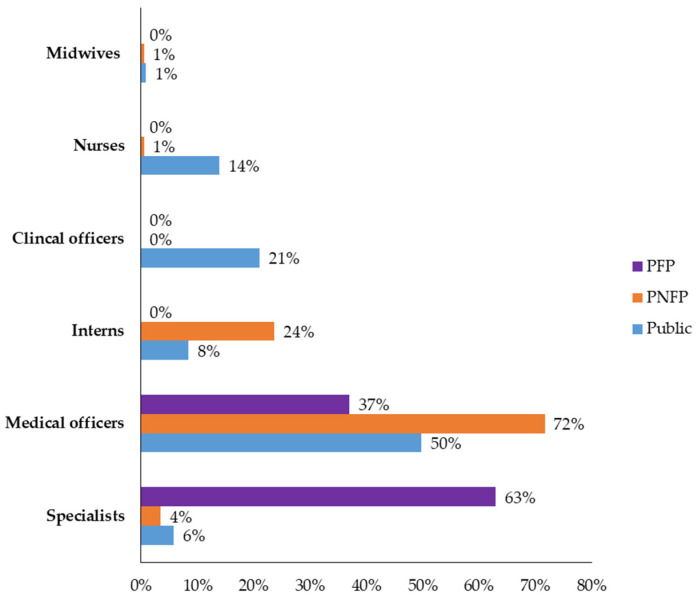
Prescriber categories at the nine health facilities (N = 882).

**Figure 2 antibiotics-10-00779-f002:**
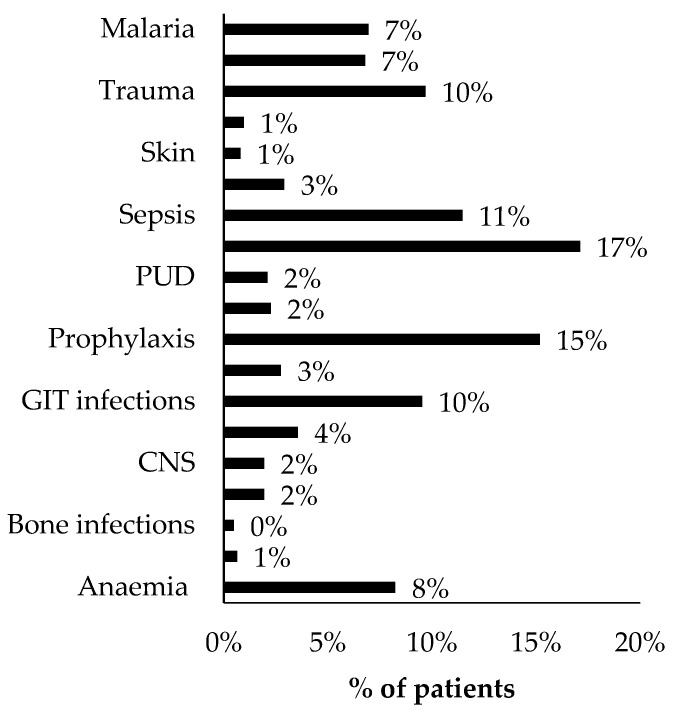
Indications for which ceftriaxone was prescribed in the nine health facilities (*n* = 622). UTIs—urinary tract infections, STIs—sexually transmitted infections, RTIs—respiratory tract infections, PUD—peptic ulcer diseases, PTB—pulmonary tuberculosis, CNS—central nervous system conditions.

**Figure 3 antibiotics-10-00779-f003:**
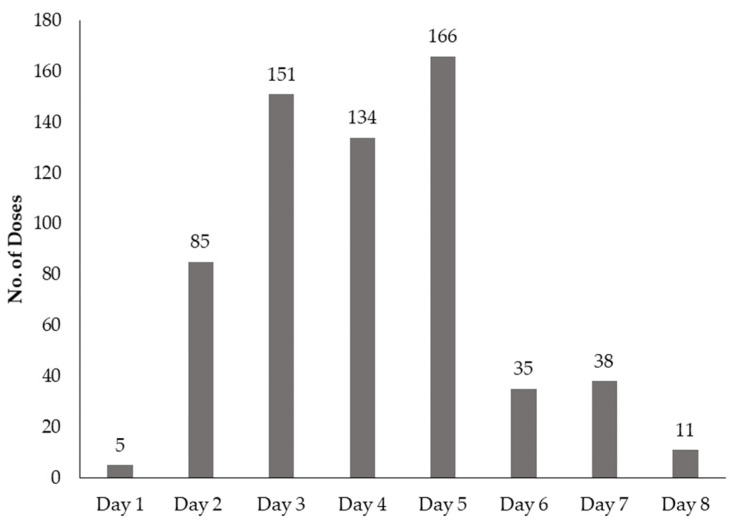
Doses of ceftriaxone that were missed during the administration of prescribed doses.

**Figure 4 antibiotics-10-00779-f004:**
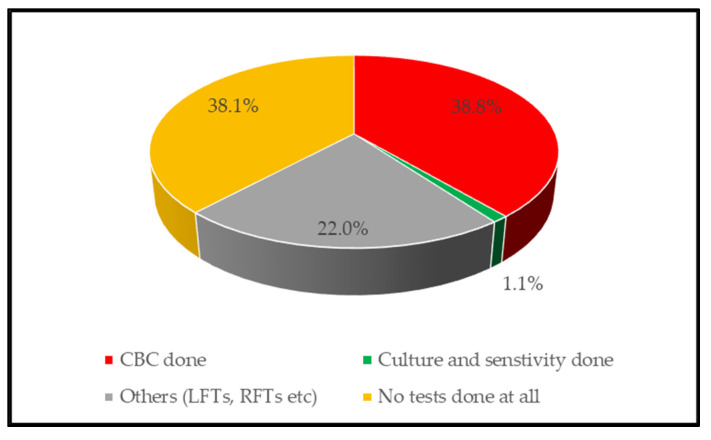
Laboratory testing rates for the nine health facilities (*n* = 885).

**Figure 5 antibiotics-10-00779-f005:**
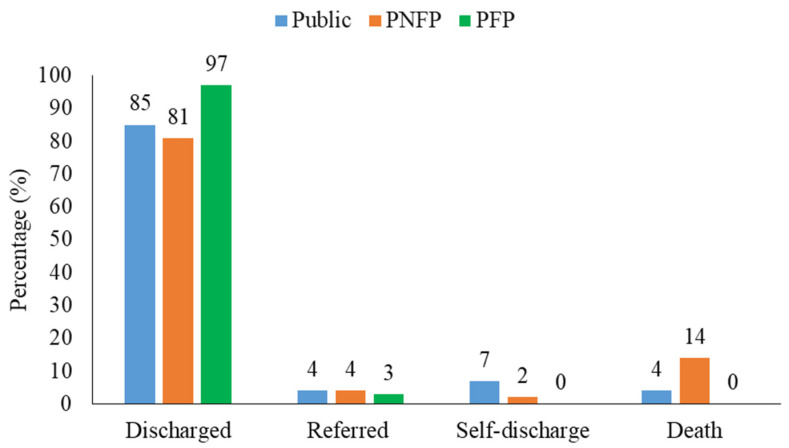
Patient treatment outcome proportions based on health facility category.

**Table 1 antibiotics-10-00779-t001:** Prescriber categories for the prescriptions reviewed (*n* = 882).

Prescriber Category	Number of Prescriptions	Percentage
Medical officers	470	53.3
Clinical officers	123	13.9
Specialists	104	11.8
Medical interns	96	10.9
Nurses	82	9.3
Midwives	7	0.8

**Table 2 antibiotics-10-00779-t002:** Appropriateness of ceftriaxone utilization based on health facility type.

Drug Use Indicators	Average Percentage Score (± SD)	Overall Average Percentage Score (± SD)	WHO Target (%) *
Appropriateness of indication			
PNFP	85.5 (2.0)		
PFP	97.0 (0.0)	83.3 (5.6)	90
Public	80.5 (2.4)		
Appropriateness of dose			
PNFP	89.5 (0.5)		
PFP	92.0 (0.0)	88.8 (3.2)	95
Public	88.0 (3.6)		
Appropriateness of duration			
PNFP	86.5 (1.5)		
PFP	90.0 (0.0)	84.1 (5.7)	95
Public	82.3 (6.1)		
Generic prescribing			
PNFP	66.9 (33.2)		
PFP	87.0 (0.0)	87.1 (21.7)	100
Public	93.5 (12.7)		
Appropriateness of testing			
PNFP	96.5 (0.65)		
PFP	77.0 (0.0)	62.2 (28.4)	100
Public	48.3 (24.2)		
Drug interactions			
PNFP	3.5 (2.5)		
PFP	2.0 (0.0)	4.1 (4.1)	0
Public	4.6 (4.9)		
Appropriateness of dispensing			
PNFP	30.9 (3.5)		
PFP	96.0 (0.0)	58.2 (26.3)	95
Public	61.0 (23.1)		

PNFP—private not-for-profit, PFP—private for profit; * WHO thresholds/targets [31].

**Table 3 antibiotics-10-00779-t003:** Completion proportions of the different doses of ceftriaxone prescribed.

Dose Completion	Ceftriaxone Doses *n* (%)				
	250 mg	500 mg	1 g	2 g	Other
Prescribed dose not completed	6 (20.0%)	25 (34.3%)	88 (34.5%)	158 (35.7%)	45 (53.6%)
Prescribed dose completed	23 (76.7%)	39 (53.4%)	161 (63.1%)	258 (58.2%)	35 (41.7%)
Unspecified dose duration	1 (3.3%)	9 (12.3%)	6 (2.4%)	27 (6.1%)	4 (4.8%)
Total	30	73	255	443	84

**Table 4 antibiotics-10-00779-t004:** Ceftriaxone doses prescribed by health workers in the different categories of health facilities.

Category	Ceftriaxone Dose Prescribed n (%)				
	250 mg	500 mg	1 g	2 g	other
Facility type					
Public facilities (PF)	26 (4.4%)	55 (9.4%)	156 (25.6%)	285 (48.6%)	65 (11.1%)
Private not-for-profit (PNFP)	4 (2.0%)	15 (7.6%)	72 (36.4%)	89 (44.9%)	18 (9.1%)
Private for-profit (PFP)	0 (0%)	3 (3%)	27 (27%)	69 (69%)	1 (1%)
Level of care					
General hospital (GH)	2 (1.0%)	11 (6.6%)	30 (20.9%)	42 (55.6%)	13 (15.8%)
Regional referral Hospital (RRH)	5 (1.1%)	24 (4.5%)	159 (29.6%)	305 (56.7%)	45 (8.4%)
Health center IV (HCIV)	23 (9.2%)	38 (15.3%)	66 (26.5%)	96 (38.6)	26 (10.4)

The number of prescriptions (N) for the different categories are PF-587, PNFP-198, PFP-100, GH-538, RRH-98, and HCIV-249.

**Table 5 antibiotics-10-00779-t005:** The doses of ceftriaxone prescribed by the different categories of prescribers (*n* = 882).

Prescriber	Ceftriaxone Dose Prescribed *n* (%)				
	250 mg	500 mg	1 g	2 g	Other
Unspecified prescriber	0 (0%)	0 (0%)	0 (0%)	1 (100%)	0 (0%)
Intern	1 (1.0%)	6 (6.3%)	26 (27.1%)	50 (52.1%)	13 (13.5%)
Medical officer	12 (2.6%)	36 (7.7%)	118 (25.1%)	269 (57.2%)	35 (7.5%)
Specialist	1 (1.0%)	4 (3.9%)	33 (31.7%)	58 (55.8%)	8 (7.7%)
Clinical officer	6 (4.9%)	6 (4.9%)	45 (36.6%)	54 (43.9%)	12 (9.8%)
Nurse	9 (11.0%)	19 (23.2%)	30 (36.6%)	9 (11.0%)	15 (18.3%)
Midwife	0 (0%)	2 (33.3%)	1 (16.7%)	2 (33.3%)	1 (16.9%)

**Table 6 antibiotics-10-00779-t006:** Factors associated with appropriate use of ceftriaxone using bivariate and multivariate logistic regression.

Variables	Appropriateness		Crude Analysis		Adjusted Analysis	
	No	Yes	Odds Ratio (CI)	*p*-Value	Adj. Odds Ratio (CI)	*p*-Value
Gender						
Male	299 (73.8%)	106 (26.2%)				
Female	398 (82.9%)	82 (17.1%)	0.6 (0.4–0.8)	0.001	0.2 (0.1–1.0)	0.045
Pregnancy						
Not pregnant	263 (79.2%)	69 (20.8%)				
Pregnant	132 (89.2%)	16 (10.8%)	0.5 (0.3–0.8)	0.009	0.4 (0.2–0.8)	0.006
Days of hospitalization						
1–3 days	312 (83.9%)	60 (16.1%)				
4–7 days	307 (76.2%)	96 (23.8%)	1.6 (1.1–2.3)	0.008	2.0 (1.2–2.5)	0.002
Above 7 days	74 (69.8%)	32 (30.2%)	2.3 (1.4–3.7)	0.001	2.2 (1.2–3.4)	0.010
Level of care						
Regional referral	183 (93.4%)	13 (6.6%)				
General hospital	312 (70.9%)	128 (29.1%)	5.8 (3.2–10.5)	0.000	3.4 (1.6–7.3)	0.002
HC IV	202 (81.1%)	47 (18.9%)	3.3 (1.7–6.2)	0.000	3.6 (1.7–7.5)	0.001
Facility type						
Public	496 (84.5%)	91 (15.5%)				
PNFP	164 (82.8%)	34 (17.2%)	1.1 (0.73–1.7)	0.579		
PFP	37 (37%)	63 (63%)	9.3 (5.8–14.8)	0.000	7.4 (3.5–15.8)	0.000
Prescriber category						
Medical intern	93 (96.9%)	3 (3.1%)				
Medical officer	370 (78.6%)	101 (21.4%)	8.5 (2.6–27.3)	0.000	4.8 (1.5–16)	0.010
Specialist	62 (59.6%)	42 (40.4%)	21.0 (6.2–70.7)	0.000	3.6 (0.9–14.0)	0.063
Clinical officer	97 (78.9%)	26 (21.1%)	8.3 (2.4–28.4)	0.001	4.3 (1.1–16.1)	0.033
Nurse	68 (82.9%)	14 (17.1%)	6.4 (1.8–23.1)	0.005	4.1 (1.0–16.8)	0.051
Midwife	4 (66.7%)	2 (33.3%)	15.5 (2.0–120.4)	0.009	8.9 (1.1–75.0)	0.044

## Data Availability

All data generated or analyzed during this study have been included in this article.

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
