# Peer review of "Evaluation of the Clinical Use of Ceftriaxone among In-Patients in Selected Health Facilities in Uganda"

_antibiotics, 2021, doi:10.3390/antibiotics10070779_

Round 1

Reviewer 1 Report

The article under review is of interest to me as it reports on the situation of the use of an antibiotic, ceftriaxone, in the health care system of an African country. It is interesting that the causes of drug misadministration may be some that in developed countries would be difficult to understand (such as forgetting to administer the drug). 
Given that there are few studies like the one presented, I would accept the paper. I would recommend a thorough review of the literature for more recent citations, as well as improvement of the tables and graphs.

Reviewer 2 Report

I read with great interest the paper. Authors wrote in my opinion important paper on AMR in a low resource setting. I believe that the question research and the setting are very important and the article high quality

Below my suggestions:

  1. Introduction: introduce better the role of surgical infection in Africa. for example, There is also high burden of maternal mortality due to infection after caesarean section and add other items (see and cite Maternal caesarean section infection (MACSI) in Sierra Leone: a case-control study. Epidemiol Infect. 2020 Feb 27;148:e40. ). Furthermore, add global burden of antimicrobial resistance with focus in africa
  2. methods and result: very good
  3. Discussion : discuss better the role of lack microbiology in low setting. Antimicrobial resitance is strongly related with microbiological identification of bacteria. Reinforce this concept. Furthermore, discuss the role of medical education during medical degree on items of AMR. In addiction, add why correct AMR is crucial also for gloibal burden of mortality due to maternal infection and other vulnerable patients (see also previous citation suggested). 
  4. In conclusion, give some public health action that come from your very interesting paper

Reviewer 3 Report

The article presents interesting results and could be of interest to a large audience, giving the increasing problem of antibiotic resistance and the need for extending the lifetime of efficient antibiotics.

The authors show that over 85.6 % of the patients were successfully discharged as having completely been treated.

However,a more extended presentation of microbiology results could increase the paper value. For example, it would be very interesting to know which was the microbial etiology of infections for which CRO was efficient versus the cases in which CRO couldn't eradicate the infection. Data referring to the antibiotic susceptibility profiles (percentage of CRO susceptible and resistant) of the microbial strains isolated from infections fro which CRO was recommended will be also very helpful, if available.

Round 2

Reviewer 3 Report

The authors have appropriately answered the reviewer' concerns.